# Early Inactivation of Membrane Estrogen Receptor Alpha (ERα) Recapitulates the Endothelial Dysfunction of Aged Mouse Resistance Arteries

**DOI:** 10.3390/ijms23052862

**Published:** 2022-03-05

**Authors:** Julie Favre, Emilie Vessieres, Anne-Laure Guihot, Linda Grimaud, Coralyne Proux, Laurent Loufrani, Françoise Lenfant, Coralie Fontaine, Jean-François Arnal, Daniel Henrion

**Affiliations:** 1MitoVasc Department, Team 2 (CarMe) and CARFI Facility, Angers University, F-49045 Angers, France; julie.favre22@gmail.com (J.F.); emilie.vessieres@univ-angers.fr (E.V.); anne-laure.guihot@univ-angers.fr (A.-L.G.); linda.grimaud@univ-angers.fr (L.G.); coralyne.proux@univ-angers.fr (C.P.); laurent.loufrani@inserm.fr (L.L.); 2Centre National de la Recherche Scientifique (CNRS), UMR 6015, F-49045 Angers, France; 3Institut National de la Santé et de la Recherche Médicale (INSERM), U1083, F-49045 Angers, France; 4Institut National de la Santé et de la Recherche Médicale (INSERM), UMR S 1121, Centre de Recherche en Biomédecine de Strasbourg (CRBS), F-67084 Strasbourg, France; 5Institut National de la Santé et de la Recherche Médicale (INSERM), U1297, Institut des Maladies Métaboliques et Cardiovasculaires (I2MC), Université Toulouse III Paul Sabatier, CHU Rangueil (University Hospital) de Toulouse, F-31432 Toulouse, France; francoise.lenfant@inserm.fr (F.L.); coralie.fontaine@inserm.fr (C.F.); jean-francois.arnal@inserm.fr (J.-F.A.); 6Angers University Hospital (CHU), F-49000 Angers, France

**Keywords:** estrogen receptors, shear stress, flow-mediated dilation, resistance arteries, ageing, endothelium

## Abstract

Flow-mediated dilation (FMD) of resistance arteries is essential for tissue perfusion but it decreases with ageing. As estrogen receptor alpha (Erα encoded by *Esr1*), and more precisely membrane ERα, plays an important role in FMD in young mice in a ligand-independent fashion, we evaluated its influence on this arteriolar function in ageing. We first confirmed that in young (6-month-old) mice, FMD of mesenteric resistance arteries was reduced in *Esr1−/−* (lacking ERα) and C451A-ERα (lacking membrane ERα). In old (24-month-old) mice, FMD was reduced in WT mice compared to young mice, whereas it was not further decreased in *Esr1−/−* and C451A-ERα mice. Markers of oxidative stress were similarly increased in old WT and C451A-ERα mice. Reduction in oxidative stress with superoxide dismutase plus catalase or Mito-tempo, which reduces mitochondrial superoxide restored FMD to a normal control level in young C451A-ERα mice as well as in old WT mice and old C451A-ERα mice. Estradiol-mediated dilation was absent in old WT mice. We conclude that oxidative stress is a key event in the decline of FMD, and that an early defect in membrane ERα recapitulates phenotypically and functionally ageing of these resistance arteries. The loss of this function could take part in vascular ageing.

## 1. Introduction

Resistance arteries are the small blood vessels located upstream of capillaries. They control blood delivery to tissues at relevant flow and pressure. Disorders in their structure and function raise capillary pressure, which exacerbates organ damage, favored by age-associated cardio- and cerebrovascular risk factors. The basal tone of these small arteries allows a tight control of local blood flow. It results in part from the interaction between pressure-induced smooth muscle contraction (myogenic tone) and flow-mediated dilation (FMD) due to the activation of endothelial cells by shear stress. A reduced FMD is the hallmark of endothelium dysfunction and FMD is altered very early in cardiovascular and metabolic disorders [1,2,3]. FMD is also progressively reduced in ageing thus amplifying the negative impact of the other risk factors [4,5,6,7]. 

Both sex and age are independent risk factors for the reduction in endothelial function [8]. Endothelial function decreases with age in healthy men and women with very low cardiovascular risk [4,5]. An age-dependent reduction in FMD is also observed in coronary human resistance arteries with a shift of the mediators involved in FMD from prostacyclin to NO with ageing whereas H_2_O_2_ is the mediator of FMD in patients with coronary artery disease [9]. We recently reported that endothelial ERα contributed to optimize flow (shear stress)-mediated dilation in young healthy mouse resistance arteries [10]. Interestingly, the only known mutation in the gene encoding ERα has been described in one man [11] and the main observed vascular defect was a strong reduction in FMD [12].

ERα belongs to the nuclear receptor superfamily and acts classically as a transcription factor, but it can also exert extranuclear, non-genomic actions by activating rapid membrane-initiated steroid signaling (MISS), as demonstrated specifically in the endothelium [13]. The vascular effects of ERα are mediated by both membrane-associated ERα, mainly through the production of NO by endothelial cells [14], and by the nuclear effects of ERα through the activating function AF2 allowing protection against atherosclerosis and hypertension [15]. Although most of the vascular protective effects of ERα are mediated by AF2-dependent nuclear effects, we have previously shown that FMD is facilitated by membrane-associated ERα-signal transduction in young male and female mice in a ligand-independent mode [10].

Beside the decline in estrogens at menopause, abnormalities in the expression and/or function of ERs in tissues, and particularly in arteries, could contribute to the failure of classic estrogens to protect arteries during ageing [16]. Thus, we investigated FMD in resistance arteries in old male mice (24 months old) in comparison to young mice (6 months old) in two mouse models: (i) totally deficient in ERα (*Esr1−/−* mice) and (ii) lacking the plasma membrane ERα as the codon for palmitoylable cystein (Cys) 451 of ERα was mutated into alanine (C451A-ERα mice) [14]. 

## 2. Results

### 2.1. Mice Age, Heart Weight, Body Weight, Blood Pressure and Heart Rate

The average age of the two groups of mice, all genotypes together, was 5.97 ± 0.12 months (n = 38 young mice) and 23.53 ± 0.38 months (n = 56 old mice). There was no significant difference in age between the four groups of young or old mice (Figure 1A,B). Similarly, body weight was equivalent between the groups independently of age (Figure 1C,D). The ratio of the left ventricle to the tibia length was not significantly affected by the genotypes or by age (Figure 1E,F). Similarly, systolic blood pressure and heart rate measured using plethysmography were not significantly affected by age and by the genotypes (Figure 1G–J). 

### 2.2. FMD in Mouse Mesenteric Arteries

Stepwise increases in intraluminal flow in perfused and cannulated mesenteric resistance arteries induced vasodilation (FMD, Figure 2A–D). 

As expected, FMD was significantly reduced in resistance arteries isolated from young mice lacking ERα compared to young littermate *Esr1*+/+ mice (Figure 2A). FMD was also significantly lower in old *Esr1*+/+ mice compared to young *Esr1*+/+ mice (Figure 2B versus Figure 2A), whereas ageing did not further reduce FMD in *Esr1*−/− mice (Figure 2B versus Figure 2A). 

A similar pattern was observed in C451A-ERα mice, which lack membrane-associated ERα compared to their littermate control C451-WT mice (Figure 2C,D).

### 2.3. Agonist-Mediated Endothelium-Dependent Dilation in Mouse Mesenteric Arteries

Acetylcholine-mediated dilation was not significantly affected by the total absence of ERα in *Esr1*−/− mice (Figure 3A,B) or by its absence at the membrane level in C451A-ERα mice (Figure 3C,D). 

Acetylcholine-mediated dilation was significantly reduced in old mice compared to young mice in all the study groups (Figure 3A,C compared to Figure 3B,D, respectively).

### 2.4. Smooth Muscle-Dependent Contraction in Mouse Mesenteric Arteries

Stepwise increases in intraluminal pressure induced contraction in isolated perfused mesenteric resistance arteries (Figure 4). Pressure-induced contraction or myogenic tone was not significantly affected by the genotype or by ageing in all study groups (Figure 4).

### 2.5. Wall Structure and Properties of Mouse Mesenteric Arteries

Internal diameter of mesenteric resistance arteries (Figure 5) was not significantly modified by the genotype or by ageing in all study groups. Cross-sectional compliance was attenuated in old mice compared to young mice in all the groups without significant difference between *Esr1*−/− and C451A-ERα mice and without difference between these mice and their littermate controls (Figure 6). 

### 2.6. Gene Expression Analysis of the Main Pathways Involved in FMD

In order to follow the impact of the absence of membrane ERα on mesenteric arteries in ageing, we measured the expression of genes representative of different main biological pathways which could affect the endothelial response to acute changes in flow or FMD (endothelial function, mechanosensing, oxidative stress, mitochondrial homeostasis, hormone-related genes, purinergic signaling). Considering the impact of ageing, important changes in gene expression were observed both in WT and C451A-ERα mice (Figure 7). They were evidenced by a down-regulation of genes involved in endothelial response and mechanosensing (cluster 1, green). Nevertheless, some genes related to oxidative stress were up-regulated in old mice (*p66Shc, Sod1, Sod2, Gpx1*). Interestingly, while in young mice the loss of membrane ERα did not significantly affect gene expression levels, different expression profiles were revealed with ageing between WT and C451A-ERα mice. Those included hormone response or metabolism pathways (*Esr1*, *AR, AhR, Cyp1b1, Comt*), the renin–angiotensin–aldosterone system (*Nr3c2, Ace*), mitochondrial homeostasis (*Dmnl1, Sirt1*) and endothelial response (*Sdc4, Kcnma1, Icam1*).

### 2.7. Effect of the Reduction in Oxidative Stress on FMD in Mouse Mesenteric Arteries

We then tested the acute effect of superoxide reduction with SOD plus catalase and of Mito-tempo on FMD. Incubation of mesenteric resistance arteries with SOD and catalase did not affect FMD in young C451-WT mice (Figure 8A), whereas it increased FMD in young C451A-ERα mice (Figure 8B). In 24-month-old mice, SOD and catalase improved FMD in both C451-WT and C451A-ERα mice (Figure 8C,D). A similar pattern was observed with arteries incubated with Mito-tempo which reduces ROS of mitochondrial origin (Figure 8A–D). 

These results suggest that the reduced FMD in C451A-ERα as compared to WT mice can be attributed to damages induced by oxidative stress, such as in ageing and or in the absence of membrane ERα mice. 

### 2.8. Effect of Age on Estradiol (E2)-Mediated Dilation in Mouse Mesenteric Arteries

Estradiol induced a concentration-dependent dilation of the resistance mesenteric arteries isolated from young WT mice, whereas no significant dilation was observed in old WT mice (Figure 9).

## 3. Discussion

The main finding of this study is that the membrane-located ERα has a facilitating role in flow (shear stress)-mediated dilation (FMD) in resistance arteries of male mice, in agreement with our previous work [10], and that this protective effect is lost in ageing. Indeed, FMD was reduced by approximately 30% in mice lacking membrane ERα, whereas in WT mice a similar reduction was observed only at the age of 24 months. However, in mice lacking membrane ERα FMD did not further decrease with age. Thus, the reduction in FMD due to the absence of membrane ERα could be presented as a premature vascular ageing. Furthermore, reducing oxidative stress restored FMD in both young mice lacking membrane ERα and in old WT mice. Thus, it is possible that membrane ERα reduces oxidative stress to facilitate FMD in young mice and that this effect is lost with ageing, due to a reduction in membrane ERα expression or function 

We investigated FMD in resistance arteries as they control local blood flow delivery to all tissues and disorders of these small arteries induce organ damage as seen in cardiovascular, metabolic and cerebrovascular disorders [17,18,19]. Resistance artery tone is counteracted by FMD and a reduced FMD is the hallmark of vascular disorders in a large number of diseases [20]. The present study showed a reduction in both FMD and receptor-dependent (acetylcholine) endothelium-mediated dilation in 24-month-old mice. This is in agreement with previous works on humans [8] and on animals [3,21,22] showing reduced FMD and more generally altered endothelium-dependent dilation in ageing. This marked alteration in endothelium-dependent dilation could be due to excessive oxidative stress as shown by an increase in the expression level of genes involved in oxidative stress observed in mesenteric resistance arteries isolated from WT and C451A-ERα mice (Figure 7), in agreement with previous studies [3,23,24]. In addition, myogenic tone was reduced in old mice as previously shown in male and female mice mesenteric arteries [25] and in male rat coronary [26] and skeletal muscle arteries [27]. Ageing was also associated with a reduced arterial compliance suggesting a change in wall structure. Indeed, arterial stiffening has been demonstrated in mesenteric resistance arteries in 2-year-old male mice [28] and it is well demonstrated in human ageing [29,30]. 

In agreement with our previous study [10], the present work confirmed the importance of the membrane-associated ERα in FMD. This effect also agrees with a single observation in one young man (31 years old). The lack of functional ERα in this man [11] was associated with a selective reduction in acute FMD [12]. Furthermore, estrogens through the nuclear AF2 function of ERα, were shown to play a key role in flow-mediated outward remodeling [31,32,33,34]. This latter is a chronic adaptation of the arterial wall leading to an increase in lumen diameter and wall mass in response to a chronic rise in blood flow in vivo [3,35]. This remodeling is observed in collateral arteries growth in ischemic diseases [35].

The vascular protection provided by membrane-associated ERα involves two arms, with a ligand-dependent pathway leading to acute NO-dependent vasodilation and involved in endothelial repair [14,36] and a ligand-independent pathway potentiating the acute response to flow or FMD (present study and [10]). By contrast, the protective effect of E2 on FMD described in pathological conditions or in menopaused women involves the ligand-dependent pathway and mainly the nuclear functions of ERα. For example, FMD, reduced in post-menopausal women, is improved by a chronic treatment with estradiol (E2) or SERMs [37,38]. Similarly, in old female rats, FMD is reduced and can be improved by estrogen supplementation [39]. On the other hand, in healthy conditions, FMD is not different between men and women and E2 does not influence FMD in healthy conditions [40]. Despite the absence of ERα, we found no significant change in body weight, left ventricle weight, systolic blood pressure and heart rate in mice, in agreement with our previous work [15]. Nevertheless, these mice develop a greater hypertension when perfused with angiotensin II [15]. Similarly, the young man with a disruptive mutation in ERα was normotensive but more susceptible to atherosclerosis [41]

Besides membrane-located ERα, G-protein coupled estrogen receptor (GPER) is also involved in the acute response to estrogens [42,43]. Although, our previous work demonstrating the role of membrane ERα in FMD in young mice has excluded a possible role for GPER [10], its involvement in the reduction in FMD observed in ageing remains to be investigated. Indeed, GPER is associated with ageing of the cardiovascular system [44] with a role in endothelial ageing [45,46]. Thus, further investigation is needed to define the role of GPER in flow-dependent signaling in vascular ageing. 

The risk of cardiovascular diseases is higher in men than in women, and the protection due to estrogen in women is progressively lost after menopause. Estrogen substitution therapy has proved to be efficient in reducing this risk although caution should be taken [13,47]. In the present study, in agreement with our previous work [10], we identified a new pathway protecting the vascular tree through the involvement of non-nuclear or membrane-associated ERα in FMD. Thus, estrogen and ERα would have a dual beneficial effect on the endothelium through the activation of eNOS expression level and NO production by E2 [13] and through a reduction in ROS production in a ligand-independent mode (present study and [10]). Noteworthy, both are reduced with ageing. E2 level decreases after menopause and flow-mediated ERα-dependent reduction in ROS production is also lost in ageing (present study). Nevertheless, although we can envision a weak impact of the former effect in our study on males, the precise pathway involved in this latter effect remains to be better defined. Indeed, this question is difficult to address as membrane ERα represents a very small percentage of the total ERα and it is barely observable using immunohistochemistry [13]. Membrane ERα is anchored to the plasma membrane through palmitoylation of its cysteine in position 451 in the mouse [14] and a post-translational dysregulation of cysteine involved in palmitoylation has been reported in ageing [48]. We have previously shown that E2-dependent acute vasodilation is mediated by membrane ERα through activation of NO production [14]. Interestingly, E2-mediated dilation was lost in old WT mice suggesting a loss of functional membrane ERα. This observation agrees with the reduced FMD found in old WT mice, which would thus be equivalent to the C451A-ERα mice which lack membrane ERα. In agreement, endothelial ageing has been shown to be associated with reduced NO production due to decreased eNOS expression and to decreased availability of L-arginine and tetrahydrobiopterin, the cofactor of eNOS [49].

The selective reduction in FMD observed in young C451A-ERα mice could represent a premature ageing of the endothelium. In agreement, FMD is early and selectively reduced in ageing in association with reduced NO bioavailability due to excessive ROS production [22,50]. Interestingly, we found that several genes involved in endothelial response and mechanosensing were down-regulated during ageing, whereas genes mainly related to oxidative stress were up-regulated in the mesenteric arteries of old mice. These observations confirm previous works [51,52]. While no differences in gene expression were found in young mice groups, it is worth noting a limitation of age-related down-regulation of several genes in old C451A-ERα mice (Figure 7, see orange and blue boxes). Thus, we may hypothesize that the functional loss of membrane-located ERα in old WT mice further affected the expression level of genes that could potentially worsen the endothelial response to flow such as genes of the renin–angiotensin–aldosterone system and of mitochondrial homeostasis (Figure 7). In agreement, we observed that Mito-tempo which reduces ROS produced by the mitochondria improved FMD in mature and old C451A-ERα mice as well as in old WT mice. Thus, mitochondrial ROS could be involved in the reduction in FMD, in agreement with previous works showing a role of these organelles in the production of ROS in ageing associated with cardiovascular disorders [53]. The pathway linking membrane-located ERα to a reduction in mitochondria-dependent ROS production remains to be further investigated. Nevertheless, mitochondrial fusion and fission are sensitive to shear stress in cultured human and bovine endothelial cells [54] and shear stress-dependent Ca^2+^ mobilization in human endothelial cells relies on mitochondria-dependent activation of endoplasmic reticulum channels [55]. In cerebral endothelial cells, activation of ERα induces a decrease in mitochondrial ROS production, possibly through up-regulation of cytochrome C activity [56]. 

Thus, targeting the pathway activated by membrane ERα in response to flow could be an attractive way to reduce oxidative damages in both healthy and diseased ageing through a reduction in ROS production and restoration of an efficient FMD in resistance arteries. Nevertheless, targeting membrane ERα per se could be useless as our results suggest that it is absent, or at least functionally deficient, in old mice.

In conclusion, our findings confirm that membrane-located ERα signaling takes part in FMD through a reduction in oxidative stress thus facilitating NO-dependent dilation. In addition, the present work suggests that this protective effect of membrane-located ERα could be lost in ageing and further suggests that an early alteration in this pathway may represent premature vascular ageing as observed in young mice lacking membrane-located ERα with a reduction in FMD equivalent to that seen in old mice.

## 4. Materials and Methods

### 4.1. Animal Protocol

We used 6-month-old and 24-month-old mice lacking the gene encoding for ERα (*Esr1*−/− compared to *Esr1*+/+ mice) [57] and mice in which the codon encoding the palmitoylable cystein (Cys) 451 of ERα was mutated into alanine (C451A-ERα mice compared to C451-WT mice) [14]. Littermate mice were used as control (wild-type, WT or +/+) in each group.

As previously described [15], systolic blood pressure (SBP) was measured on conscious mice using a non-invasive and fully automated and computerized tail-cuff method (photoplethysmograph BP-2000 Blood Pressure Analysis System™, Visitech Systems, Apex, NC, USA). The means of 5-day measurements were computed after one-week adaptation period.

Mice were euthanized using a CO_2_ chamber and the mesentery was quickly removed and placed in ice-cold physiological salt solution (PSS). Several segments of mesenteric resistance arteries were isolated for the functional and biochemical studies.

The experiment complied with the European Community standards on the care and use of laboratory animals and the Guide for the Care and Use of Laboratory Animals published by the US National Institutes of Health (NIH Publication No. 85–23, revised 1996). The protocol was approved by the regional ethics committee (“Comité d’éthique en Expérimentation Animale des Pays de la Loire”, authorization # CEEA PdL 2012.141).

### 4.2. Flow-Mediated Dilation in Mesenteric Arteries In Vitro

Arterial segments, approximately 200 µm in internal diameter, were cannulated at both ends on glass micro-cannulae and mounted in a video-monitored perfusion system (Living System, LSI, Burlington, VT, USA) [58]. The arterial segment was bathed in a 5 mL organ bath containing a physiological salt solution (PSS, pH: 7.4, pO_2_:160 mmHg and pCO_2_: 37 mmHg) [59]. Perfusion of the artery was obtained with 2 peristaltic pumps, one controlling flow and one under the control of a pressure-servo control system (LSI, Burlington, VT, USA) allowing the control of pressure [59]. Pressure at both ends of the arterial segment was monitored using pressure transducers (MP-4 system, LSI, Burlington, VT, USA). To measure flow-mediated dilation (FMD), pressure was set at 75 mmHg and arterial tone was increased with phenylephrine (1 µmol/L). Flow (3 to 50 µL per min) was then generated through the distal pipette with a peristaltic pump [59]. 

In separate series of experiments, FMD was measured before and after incubation (20 min) of the arteries with superoxide dismutase (SOD, 120 U/mL) plus catalase (80 U/mL) [60] or Mito-Tempo (1 µmol/L) [61].

Other segments of mesenteric arteries were used for a cumulative concentration–response curve (CRC) to acetylcholine (10^−9^ to 10^−5^ mol/L) or to estradiol (E2, 10^−9^ to 10^−7^ mol/L) after precontraction of the arterial segment with phenylephrine (1µmol/L) to contract the arteries by approximately 50%. 

Myogenic tone (MT) was determined in response to stepwise increases in intra luminal pressure from 10 to 125 mmHg using a video-monitored perfusion system as described above. MT at a given perfusion pressure was defined as the magnitude of the percent myogenic tone (%MT) at that pressure. The %MT was expressed by the active (AD) and passive vessel diameters (PD) such that %MT = [(PD−AD)/PD]·100% [62].

At the end of each experiment, arteries were bathed in a Ca^2+^-free PSS containing ethylene-bis-(oxyethylenenitrolo) tetra-acetic acid (2 mmol/L) and sodium nitroprusside (10 µmol/L). Pressure was then increased in steps from 10 to 125 mmHg, in the absence of flow, to determine passive arterial diameter and passive mechanical properties of the arterial wall, as previously described [60]. 

### 4.3. Quantitative Real-Time PCR

Gene expression was investigated using quantitative polymerase chain reaction after reverse transcription of total RNA (RT-qPCR). Mesenteric arteries were stored at −20 °C in RNAlater Stabilization Reagent (Qiagen, Valencia, CA, USA) until use. RNA was extracted using the RNeasy^®^ Micro Kit (Qiagen, Valencia, CA, USA) following the manufacturer’s instructions. RNA extracted (300 ng) was used to synthesize cDNA using the QuantiTect^®^ Reverse Transcription Kit (Qiagen, Valencia, CA, USA). RT-qPCR was performed with Sybr^®^ Select Master Mix (Applied Biosystems Inc., Lincoln, CA, USA) reagent using a LightCycler 480 Real-Time PCR System (Roche, Branchburg, NJ, USA). Primer sequences are shown in Table 1. *Hprt*, *ActB* and *Gusb* were used as housekeeping genes. Analysis was not performed when Ct values exceeded 35. Results were expressed as: 2(Ct target-Ct housekeeping gene) and heatmaps of expression values were all generated using MeV. Expression values shown within the heatmaps were normalized per mRNA as fold changes of means of young C451-WT mice. 

### 4.4. Statistical Analysis

Results were expressed as means ± SEM. Significance of the differences between groups was determined by analysis of variance (two-way ANOVA for consecutive measurements for pressure–diameter curves, myogenic, FMD and CRC to ACh) followed by a Bonferroni test. The Mann–Whitney test was used for the other comparisons. Probability values less than 0.05 were considered significant. 

## Figures and Tables

**Figure 1 ijms-23-02862-f001:**
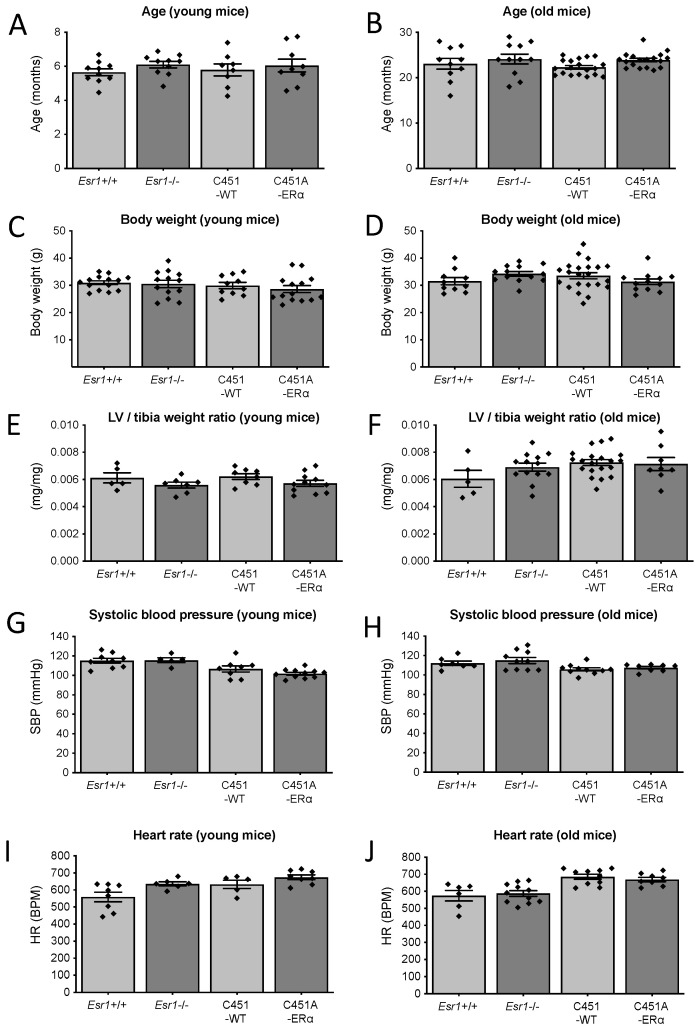
Biometric characteristics of the *Esr1*−/− and C451A-ERα mice. Age (**A**,**B**), body weight (**C**,**D**), the ratio left ventricle (LV) weight/tibia length (**E**,**F**), systolic blood pressure (SBP, **G**,**H**) and heart rate (HR, **I**,**J**) were measured in young mice (6 months old, **A**,**C**,**E**,**G**,**I**) and old mice (24 months old, **B**,**D**,**F**,**H**,**J**) *Esr1*−/− and C451A-ERα mice and their littermate controls (*Esr1*+/+ or C451-WT). Mean ± SEM is shown (n = 5 to 18 per group). BPM: beats per minute. NS, Mann–Whitney test.

**Figure 2 ijms-23-02862-f002:**
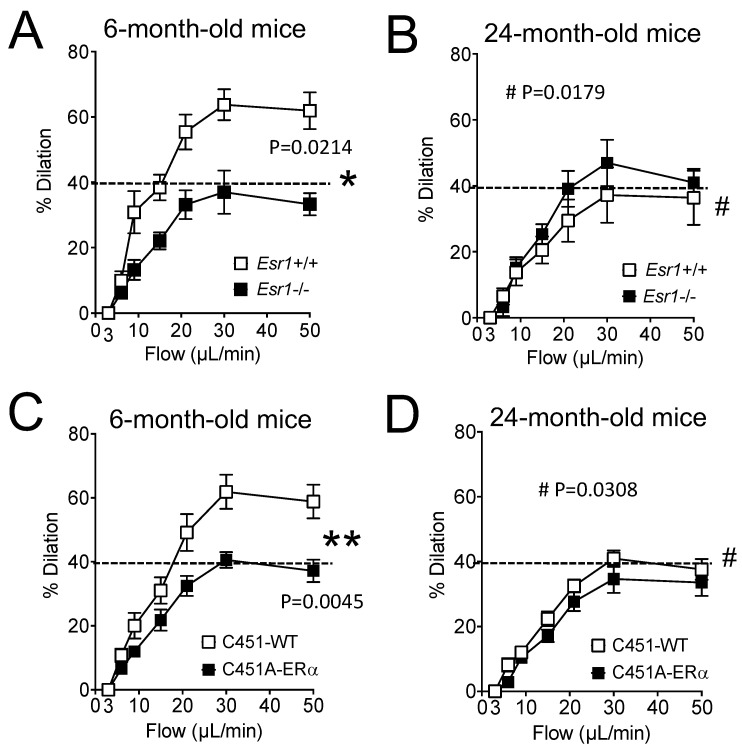
Flow-mediated dilation in *Esr1*−/− and C451A-ERα mice. Flow-mediated dilation was determined in mesenteric resistance arteries isolated from young (**A**,**C** 6-month-old) and old (**B**,**D** 24-month-old) *Esr1*−/− (**A**,**B**) and C451A-ERα (**C**,**D**) male mice and their littermate controls (*Esr1*+/+ or WT). Mean ± SEM is shown (n = 11 *Esr1*+/+, 10 *Esr1*−/−, 7 C451-WT, 8 C451A-ERα mice). * *p* < 0.05, ** *p* < 0.01, two-way ANOVA for repeated measurements (flow), *Esr1*−/− or C451A-ERα versus the corresponding WT. # *p* < 0.05, two-way ANOVA for repeated measurements, old versus young mice within each group.

**Figure 3 ijms-23-02862-f003:**
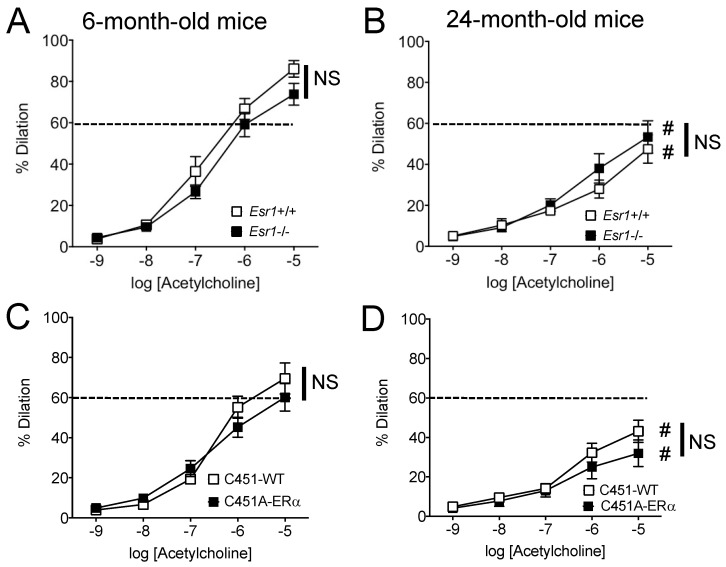
Endothelium-dependent dilation induced by acetylcholine. Acetylcholine-mediated dilation was measured in mesenteric resistance arteries isolated from young (**A**,**C** 6-month-old) and old (**B**,**D** 24-month-old) *Esr1*−/− (**A**,**B**) and C451A-ERα (**C**,**D**) male mice and their littermate controls (*Esr1*+/+ or WT). Mean ± SEM is shown (n = 11 *Esr1*+/+, 10 *Esr1*−/−, 6 C451-WT, 6 C451A-ERα mice). NS (not significant), two-way ANOVA for repeated measurements, *Esr1*−/− or C451A-ERα versus *Esr1*+/+ or WT, respectively. # *p* < 0.05, two-way ANOVA for repeated measurements, old versus young mice within each group.

**Figure 4 ijms-23-02862-f004:**
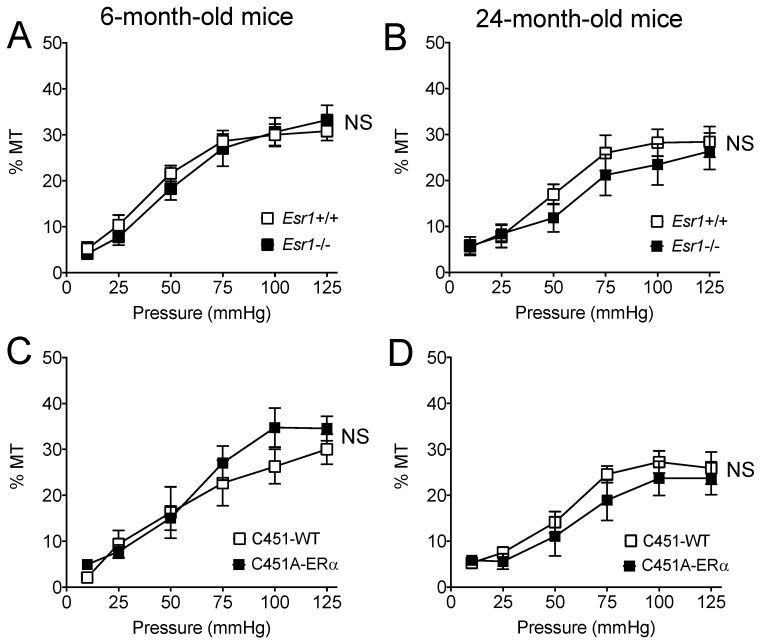
Pressure-induced myogenic tone in Esr1−/− (**A**,**B**) and C451A-ERα mice (**C**,**D**). Myogenic tone was measured in mesenteric resistance arteries isolated from young (**A**,**C** 6-month-old) and old (**B**,**D** 24-month-old) *Esr1*−/− (**A**,**B**) and C451A-ERα (**C**,**D**) male mice and their littermate controls (*Esr1*+/+ or WT). Mean ± SEM is shown (n = 7 *Esr1*+/+, 8 *Esr1*−/−, 6 C451-WT and 6 C451A-ERα mice). NS, two-way ANOVA for repeated measurements, *Esr1*−/− or C451A-ERα versus the corresponding WT group NS, two-way ANOVA for repeated measurements, old versus young.

**Figure 5 ijms-23-02862-f005:**
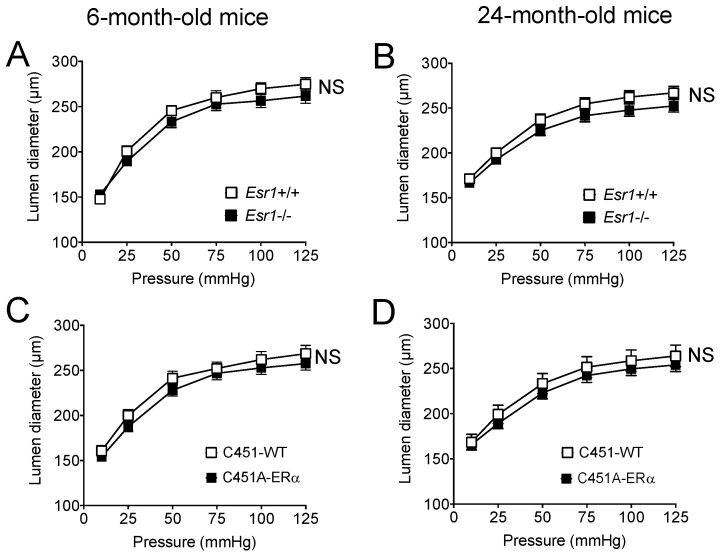
Arterial diameter in *Esr1*−/− (**A**,**B**) and C451A-ERα mice (**C**,**D**). Passive arterial diameter was measured in mesenteric resistance arteries isolated from young (**A**,**C** 6-month-old) and old (**B**,**D** 24-month-old) *Esr1*−/− (**A**,**B**) and C451A-ERα (**C**,**D**) male mice and their littermate controls (*Esr1*+/+ or WT). Mean ± SEM is shown (n = 7 *Esr1*+/+, 8 *Esr1*−/−, 6 C451-WT, 6 C451A-ERα mice). NS, two-way ANOVA for repeated measurements, *Esr1*−/− versus *Esr1*+/+ and C451A-ERα versus C451-WT NS, two-way ANOVA for repeated measurements, old versus young within each group.

**Figure 6 ijms-23-02862-f006:**
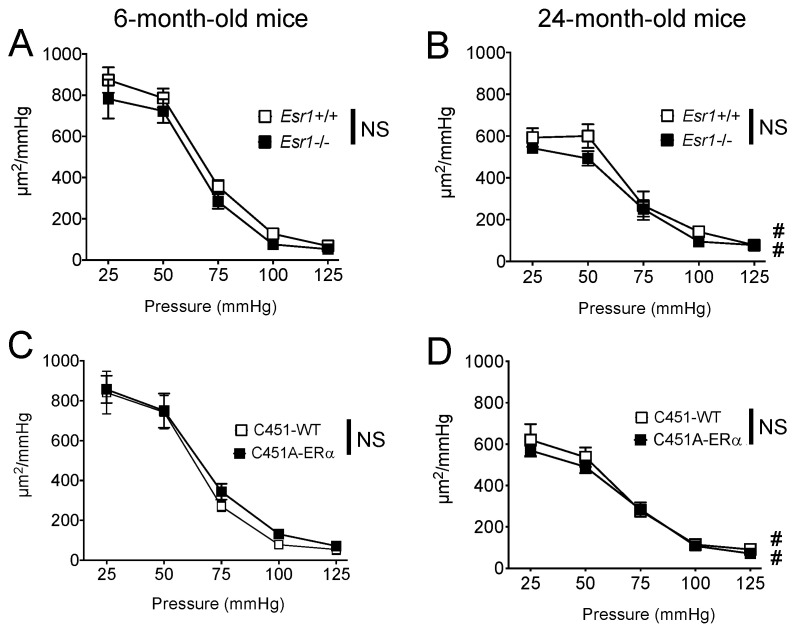
Arterial compliance in *Esr1*−/− and C451A-ERα mice. Cross-sectional arterial compliance was measured in mesenteric resistance arteries isolated from young (**A**,**C** 6-month-old) and old (**B**,**D** 24-month-old) *Esr1*−/− (**A**,**B**) and C451A-ERα (**C**,**D**) male mice and their littermate controls (*Esr1*+/+ or WT). Mean ± SEM is shown (n = 7 *Esr1*+/+, 8 *Esr1*−/−, 6 C451-WT, 6 C451A-ERα mice). NS, two-way ANOVA for repeated measurements, *Esr1*−/− or C451A-ERα versus the corresponding WT groups. # *p* < 0.05, two-way ANOVA for repeated measurements, old versus young.

**Figure 7 ijms-23-02862-f007:**
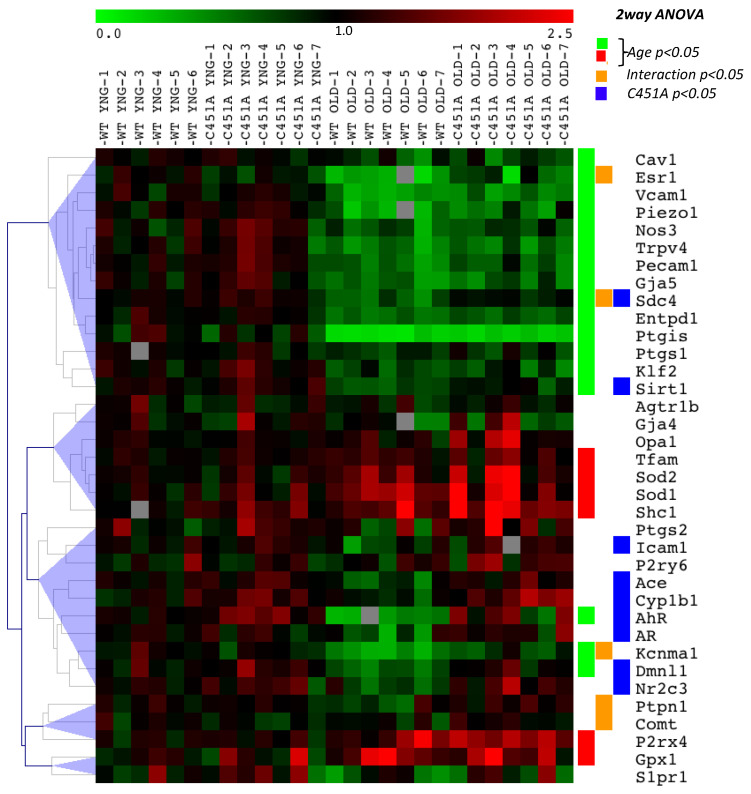
Gene expression profile in mesenteric arteries isolated from mice lacking membrane ERα.RT-qPCR gene expression analysis of mesenteric resistance arteries isolated from young (6-month-old: YNG) and old (24-month-old: OLD) C451A-ERα and C451-WT male mice (n = 6–7 mice per group). Two-way ANOVA analysis shown on the figure. Age: *p* < 0.05 YNG vs. OLD (green and red labels: down-regulated and up-regulated genes vs. YNG, respectively). C451A: *p* < 0.05 C451-WT vs. C451A-ERα (blue label). Interaction: *p* < 0.05 between Age and C451A-ERα genotype (orange label).

**Figure 8 ijms-23-02862-f008:**
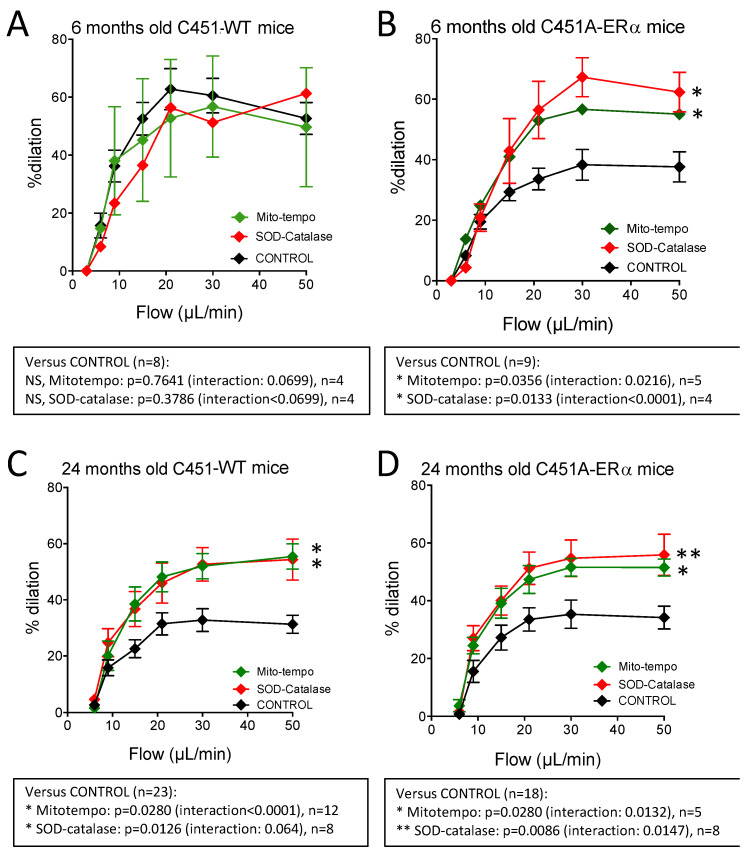
Flow-mediated dilation and oxidative stress in mice lacking membrane ERα. Flow-mediated dilation was determined in mesenteric resistance arteries isolated from male young (**A**,**B** 6-month-old) and old (**C**,**D** 24-month-old) C451A-ERα and their littermate controls C451-WT. Flow-mediated dilation was measured before and after the addition of SOD plus catalase or Mito-tempo in the physiological salt solution bathing the arterial segments (20 min incubation). Mean ± SEM is shown. Two-way ANOVA for repeated measurements, effect of SOD plus catalase or Mito-tempo. *p* values shown under each graph.

**Figure 9 ijms-23-02862-f009:**
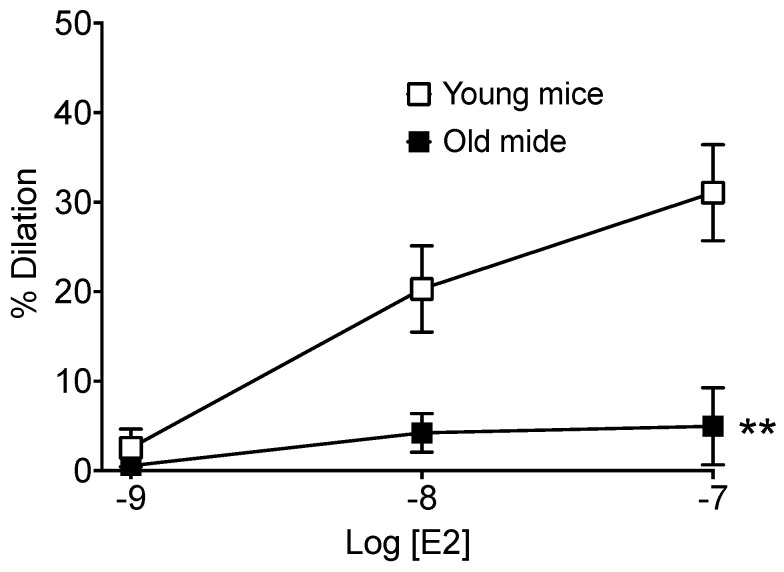
Estradiol-mediated dilation in wild-type mice. Estradiol (E2)-mediated dilation was determined in mesenteric resistance arteries isolated from young (6-month-old) and old (24-month-old) wild-type (WT) male mice. Mean ± SEM is shown (n = 8 mice per group). ** *p* = 0.0074, Two-way ANOVA for repeated measurements.

**Table 1 ijms-23-02862-t001:** Primer sequences used for the RT-qPCR.

Gene	Protein	NCBI Reference Sequence	Forward Sequence (5’–3’)	Reverse Sequence (5’–3’)
** *Ace* **	Ace	NM_009598.2	GGAACAAGTCGATGTTAGAGAAGC	ACAGAGGTACACTGCTTGATCCT
** *ActB* **	Beta-actin	NM_007393.1	GCCGGGACCTGACTGACTAC	TTCTCCTTAATGTCACGCACGAT
** *Agtr1b* **	AT1R (b)	NM_175086.3	GTGACATGATCCCCTGACAGT	AGTGAGTGAACTGTCTAGCTAAATGC
** *Ahr* **	Transcription factor cyp1b1	NM_001314027.1	GATGCCAAAGGGCAGCTTATTC	CCACCTCCAGCGACTGTGTTT
** *Ar* **	Androgen receptor	NM_013476.4	CCAGTCCCAATTGTGTCAAA	TCCCTGGTACTGTCCAAACG
** *Cav1* **	Caveolin1	NM_007616.2	AACGACGACGTGGTCAAGA	CACAGTGAAGGTGGTGAAGC
** *Comt* **	Catechol O-methyltransferase	NM_001111062.1	CCGCTACCTTCCAGACACAC	GTTCCCGGGACAATGACA
** *Cyp1b1* **	Cytochrome P450 1b1	NM_009994.1	AGCCAGGACACCCTTTCC	CCTGAACATCCGGGTATCTG
** *Dnm1l* **	Drp1	NM_001276340.1	AGATCGTCGTAGTGGGAACG	CCACTAGGCTTTCCAGCACT
** *Entpd1* **	NtPDase1	NM_009848.3	CTCCTGCAAGGCTATAACTTCAC	GCGTTGCTGTCTTTGATCTTG
** *Esr1* **	ER alpha	MN_007956.4	GCTCCTAACTTGCTCCTGGAC	CAGCAACATGTCAAAGATCTCC
** *Gja4* **	Cx37	NM_008120.3	TCCTGGGAAAAAGCACTGAT	CTGTGTCTGTCCAGGTGACG
** *Gja5* **	Cx40	NM_008121.2	CAGTGTGATCCTCCTTTTAGGG	TTTCCTGCCTCACACTCCTT
** *Gpx1* **	gPx-1	NM_008160.6	TTTCCCGTGCAATCAGTTC	TCGGACGTACTTGAGGGAAT
** *Gusb* **	GUSB	NM_010368.1	CTCTGGTGGCCTTACCTGAT	CAGTTGTTGTCACCTTCACCTC
** *Hprt ter* **	HPRT	NM_013556.2	TGATAGATCCATTCCTATGACTGTAGA	AAGACATTCTTTCCAGTTAAAGTTGAG
** *Icam1* **	ICAM	NM_010493.2	GCTACCATCACCGTGTATTCG	AGGTCCTTGCCTACTTGCTG
** *Kcnma1* **	Bkca alpha1	NM_001253358.1	GTACCTGTGGACCGTTTGCT	CGTCCACTGGCTTGAGAGTA
** *Klf2* **	KLF	NM_008452.2	CTAAAGGCGCATCTGCGTA	TAGTGGCGGGTAAGCTCGT
** *Nos3* **	eNOS	NM_008713.4	CCAGTGCCCTGCTTCATC	GCAGGGCAAGTTAGGATCAG
** *Nr2c3* **	Nuclear receptor subfamily 3, group C, member 2	NM_001083906.1	TTCGGAGAAAGAACTGTCCTG	CCCAGCTTCTTTGACTTTCG
** *Opa1* **	Opa1	NM_001199177.1	ACCAGGAGAAGTAGACTGTGTCAA	TCTTCAAATAAACGCAGAGGTG
** *P2rx4* **	P2X4	NM_011026.2	CCAACACTTCTCAGCTTGGAT	TGGTCATGATGAAGAGGGAGT
** *P2ry6* **	P2Y6	NM_183168.1	TCTTCCATCTTGCATGAGACA	GGATGGTGCCATTGTCCT
** *Pecam1* **	CD31	NM_001032378.1	CGGTGTTCAGCGAGATCC	CGACAGGATGGAAATCACAA
** *Piezo1* **	PIEZO1	NM_001037298.1	ATCAAGTGCAGCCGAGAGAC	TAATGAGGCCTCCCATACCA
** *Ptgis* **	PgI2 synthase, prostacyclin synthase	NM_008968.3	AGGAAAAGCACGGTGACATATT	CCCACACCACTGTGTCGTAA
** *Ptgs1* **	COX1	NM_008969.3	CCTCTTTCCAGGAGCTCACA	TCGATGTCACCGTACAGCTC
** *Ptgs2* **	COX2	NM_011198.3	GGGAGTCTGGAACATTGTGAA	GCACATTGTAAGTAGGTGGACTGT
** *Ptpn1* **	Protein tyrosine phosphatase, non-receptor type 1	NM_011201.3	CATCATGGAGAAAGGCTCGT	CCTGTGTCATCAAAGACCATCT
** *S1pr1* **	Sphingosine-1-phosphate receptor 1	NM_007901.5	CGGTGTAGACCCAGAGTCCT	AGCTTTCCTTGGCTGGAG
** *Sdc4* **	Syndecan4	NM_011521.2	GACCTCCTGGAAGGCAGATA	GCTCCTCCGTGTCATCCA
** *Shc1a* **	p66shc isoform a	NM_001113331	GGACCCATTCTGCCTCCTCT	GCCAGCTTCAGGTTGCTCAT
** *Sirt1* **	Sirtuin1	NM_019812.2	CAGTGAGAAAATGCTGGCCTA	TTGGTGGTACAAACAGGTATTGA
** *Sod1* **	SOD1	NM_011434.1	CAGGACCTCATTTTAATCCTCAC	TGCCCAGGTCTCCAACAT
** *Sod2* **	SOD2	NM_013671.3	GACCCATTGCAAGGAACAA	GTAGTAAGCGTGCTCCCACAC
** *Tfam* **	Tfam	NM_009360.4	CAAAGGATGATTCGGCTCAG	AAGCTGAATATATGCCTGCTTTTC
** *Trpv4* **	Trpv4	NM_022017.3	GGCAAGAGTGAAATCTACCAGTACTAT	ACCGAGGACCAACGATCC
** *Vcam1* **	VCAM1	NM_011693.2	TGATTGGGAGAGACAAAGCA	AACAACCGAATCCCCAACTT

## Data Availability

The data presented in this study are available on request from the corresponding author.

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
