# Peer review of "Early Inactivation of Membrane Estrogen Receptor Alpha (ERα) Recapitulates the Endothelial Dysfunction of Aged Mouse Resistance Arteries"

_ijms, 2022, doi:10.3390/ijms23052862_

Round 1

Reviewer 1 Report

This is a very nice paper on the role of ERalpha in endothelial function and aging.

The study is very well designed and performed, data are very interesting, important, very nicely presented and discussed.

Congratulations!

Just a small correction needed:

lane 68: please delete one ''lacking''

Author Response

Thank you very much for your comments.

We removed one “lacking” on lane 68 (now 71)

Reviewer 2 Report

In the present study the contribution of alterations in the function of membrane ERα receptors in the decline of flow-mediated dilation (FMD) in aged mouse mesenteric arteries was investigated. This study is in line with a similar study published by the same team recently (Favre et al., eLife 2021; 10e68695). The present study extends that  study to investigated how the reported influence of membrane ERα receptors on FMD is altered from 6 to 24 months old mice.

The study is well designed and well performed. It is well supported scientifically and the conclusions are according to the main results obtained.  The discussion covers the topics investigated except the data from transcriptomic analysis that should also be discussed properly.

The authors should also comment on a putative role of GPER on the responses observed. 

The manuscript is easy to read but paragraph from lines 232 up to 241 should be reformulated. In this paragrath, sentences are to long what makes the reading difficult.

Author Response

We wish to thank you for your comments. We modified the text as recommended and as detailed below:

Comment 1: The discussion covers the topics investigated except the data from transcriptomic analysis that should also be discussed properly.

Response: We added 2 paragraphs in the discussion stating that the gene analysis highlighted a role for oxidative stress in vascular aging: See lanes 316-320 and 352-361

Comment 2: The authors should also comment on a putative role of GPER on the responses observed. 

Response: We added a paragraph on GPER in the discussion (lane 350-356).

Besides membrane-located ERα, G-protein coupled estrogen receptor (GPER) is also involved in the acute response to estrogens [42, 43]. Although, our previous work demonstrating the role of membrane ERα in FMD in young mice has excluded a possible role for GPER [10], its involvement in the reduction in FMD observed in ageing remains to be investigated. Indeed, GPER is associated with aging of the cardiovascular system [44] with a role in endothelial ageing [45, 46]. Thus, further investigation is needed to define the role of GPER in flow-dependent signaling in vascular ageing

Comment 3: The manuscript is easy to read but paragraph from lines 232 up to 241 should be reformulated. In this paragraph, sentences are too long what makes the reading difficult.

Response: We have rephrased this paragraph. Now: lane 326 to lane 333:

The text is now: “In agreement with our previous study [10], the present work confirmed the importance of the membrane-associated ERα in FMD. This effect also agrees with a single observation in one young man (31 years old). The lack of functional ERα in this man [11] was associated with a selective reduction in acute FMD [12]. Furthermore, estrogens through the nuclear AF2 function of ERα, were shown to play a key role in flow-mediated outward remodeling [31-34]. This latter is a chronic adaptation of the arterial wall leading to an increase in lumen diameter and wall mass in response to a chronic rise in blood flow in vivo [3, 35]. This remodeling is observed in collateral arteries growth in ischemic diseases [35]”

Reviewer 3 Report

The manuscript is submitted by Henrion’s group in France, which published a similar line of study in 2021 using the ERa mutant mouse lines established by Lenfant’s lab (co-authors of the manuscript). A new finding reported here are that the flow-mediated dilation (FMD) in the mesenteric artery (an ex vivo model) carrying a mutation in membrane ERa in young age presents a similar profile of those in old mice, and the reduced FMD in mice regardless of the genotype and age is restored to the WT level of young mice by alleviating oxidative stress.

The manuscript is well written, and the experimental data are clear and compelling. However, there is an issue in how the gene expression data was interpreted.

Major issue

In the discussion section, the authors ought to explain possible reasons why the gene expression of young mutants does not recapitulate the one in old WT mice as the data does not support the title of the manuscript. Were there any genes that showed differential expression between young WT and C451A, especially the ones related to oxidative stress? Include that info in the result section under 2.6. Also, discuss in the discussion section how FMD may be restored with SOD-catalase or MitoTempo in the arteries of the young mutants when elevated gene expression in oxidative stress-related genes was not observed.

Minor issues

Line 68. Correct the duplication of lacking.

Figure 1F. In Results or Discussion, briefly mention why old mERa mutants do not show any consequences in the mass of the left ventricle. Although the acute model of FMD is the focus of this study, has it been characterized whether the mutant mice, either or both of ERa-/- and C451A, manifest any phenotypes of hypertension (increased afterload) from a young age? If so, describe them or cite the reference.

Line 160. Rename the running title. The use of “transcriptomic” is misleading because only a set of genes were analyzed for the expression of transcripts.

Figure 7. Replace the value, 0.9198232 with 1.0 at the corresponding point in the expression spectrum.

Line183-185. Correct the duplicated entries.

Line 246. Spell out estradiol for E2.

Author Response

We wish to thank you for your comments and suggestions. We have added new data to the manuscript and extended the discussion as described below:

Comment 1: In the discussion section, the authors ought to explain possible reasons why the gene expression of young mutants does not recapitulate the one in old WT mice as the data does not support the title of the manuscript. Were there any genes that showed differential expression between young WT and C451A, especially the ones related to oxidative stress? Include that info in the result section under 2.6. Also, discuss in the discussion section how FMD may be restored with SOD-catalase or MitoTempo in the arteries of the young mutants when elevated gene expression in oxidative stress-related genes was not observed.

Response: The main message of the gene expression analysis shown here is that oxidative stress becomes obvious in old mice irrespective of the mutation. This led us to investigate the effect of the antioxidant drugs (SOD-catalase or MitoTempo) on FMD in old mice. Second, we show that in old mice, FMD in WT mice is reduced due to oxidative stress and that FMD in C451A-ERα old mice remains reduced due to ROS production, as in young C451A-ERα mice. In agreement with our previous study (Favre et al., Elife. 2021 Nov 29;10:e68695) we found that FMD was improved by antioxidants in young C451A-ERα and not in young WT mice.

There are possibly two different processes: 1) in young mice membrane-ERα reduces ROS production when flow increases acutely so that flow-mediated NO production is more efficient. In mutant mice this membrane ERα-dependent protection is lost and consequently FMD is reduced (our first study: Favre et al., Elife. 2021 Nov 29;10:e68695). FMD is an acute response to flow and no change in gene expression is involved as shown in this previous work. This is confirmed by the present study showing no change in gene expression between WT and mutant (C451A- ERα) young mice. This observation (absence of change in gene expression in young C451A-ERα) also agrees with previous works showing that changes in gene expression are mainly associated to the nuclear effects of ERα (Physiol Rev. 2017 Jul 1;97(3):1045-1087). 2) In old mice, oxidative stress reduces the acute response to flow, and this could be due to the aging process and the associated oxidative stress as both WT and mutant mice are affected. Consequently, antioxidant drugs improved FMD in both WT and mutant mice. Alternatively, but not exclusively, the protective effect of membrane ERα could be lost in old WT mice whereas in old mutant mice, the loss of this protection remains over time. As tools allowing to see the receptors are not very specific and as membrane-located ERα only represent a small percentage of the receptors we used an alternative technique to test the presence of membrane ERα. As shown in our previous study (Proc Natl Acad Sci U S A. 2014 Jan 14;111(2): E283-90), membrane ERα mediates the acute dilatory response of estradiol and this dilation is absent in mutant (C451A- ERα) young mice. Thus, we measured estradiol (E2)-mediated dilation in young and old WT mice (new data in figure 8). We found that E2-mediated dilation was strongly reduced in old WT mice, suggesting a functional defect in the membrane Erα-associated response in old WT mice. This is a functional absence as we cannot show the receptor in endothelial cells, but this is also the function (endothelium-dependent dilation) involved in FMD. Thus, another possible explanation for the present observation is that the facilitating role of membrane ERα in FMD is lost in old WT mice. Consequently, FMD is reduced by ROS in both old WT mice and mutant mice (young and old).  

We modified the discussion as follows:

  • Role of oxidative stress in endothelium-dependent dilation in ageing:

Line 3216-320: “This marked alteration in endothelium-dependent dilation could be due to excessive oxidative stress as shown by an increase in the expression level of genes involved in oxidative stress observed in mesenteric resistance arteries isolated from WT and C451A-ERα mice (Figure 7), in agreement with previous studies [3, 23, 24] »

  • Possible reasons why the gene expression of young mutants does not recapitulate the one in old WT mice:

We modified the discussion as follows (lines 404 – 413):

« Interestingly, we found that several genes involved in endothelial response and mechanosensing were down-regulated during ageing whereas genes mainly related to oxidative stress were up-regulated in the mesenteric arteries of old mice. These observations confirm previous works [50, 51]. While no differences in gene expression were found in young mouse groups, it is worth to note a limitation of ageing-related down-regulation of several genes in old C451A-ERα mice (Figure 7, see orange and blue boxes). Thus, we may hypothezise that the functional loss of membrane-located ERα in old WT mice further affected the expression level of genes that could potentially worsen the endothelial response to flow such as genes of the renin-angiotensin-aldosterone system and of mitochondrial homeostasis (Figure 7)”

Comment 2: Line 68. Correct the duplication of lacking.

Response: We suppressed the second lacking (now line 71)

Comment 3: Figure 1F. In Results or Discussion, briefly mention why old mERa mutants do not show any consequences in the mass of the left ventricle. Although the acute model of FMD is the focus of this study, has it been characterized whether the mutant mice, either or both of ERa-/- and C451A, manifest any phenotypes of hypertension (increased afterload) from a young age? If so, describe them or cite the reference.

Response: No hypertension has been observed in the different model of mutant mice (AF2°ERα, C451A- ERα and ERα -/-) as shown in Guivarc’h et al 2018 and 2020 (reference shown below and also present in the manuscript). As these 2 studies were performed in female mice and the present work in male mice, we have added to the present manuscript blood pressure and heart rate values measured in the different groups of young and old mice. This new data is shown in figure 1 G to J. The present data also agrees with the observation made in one man with a disruptive mutation in ERα who was normotensive although more susceptible to atherosclerosis (Sudhir et al, see below). Whereas the absence of ERα has no effect on blood pressure its absence amplifies the pressure response of angiotensin II. We have shown that angiotensin II-induced hypertension in mice is greater in the absence of ERα. This protective effect of ERα is mediated by the nuclear activating function AF2 of ERα (Guivarc’h, 2020) by contrast with the protective effect of ERα in FMD which is mediated by the membrane-located receptor. We have extended this part of the discussion (lines 345-349).

References for the response to comment 3:

Guivarc'h, E.; Buscato, M.; Guihot, A. L.; Favre, J.; Vessieres, E.; Grimaud, L.; Wakim, J.; Melhem, N. J.; Zahreddine, R.; Adlanmerini, M.; Loufrani, L.; Knauf, C.; Katzenellenbogen, J. A.; Katzenellenbogen, B. S.; Foidart, J. M.; Gourdy, P.; Lenfant, F.; Arnal, J. F.; Henrion, D.; Fontaine, C., Predominant Role of Nuclear Versus Membrane Estrogen Receptor alpha in Arterial Protection: Implications for Estrogen Receptor alpha Modulation in Cardiovascular Prevention/Safety. J Am Heart Assoc 2018, 7, (13), e008950.

Guivarc'h, E.; Favre, J.; Guihot, A. L.; Vessieres, E.; Grimaud, L.; Proux, C.; Rivron, J.; Barbelivien, A.; Fassot, C.; Briet, M.; Lenfant, F.; Fontaine, C.; Loufrani, L.; Arnal, J. F.; Henrion, D., Nuclear Activation Function 2 Estrogen Receptor alpha Attenuates Arterial and Renal Alterations Due to Aging and Hypertension in Female Mice. J Am Heart Assoc 2020, 9, (5), e013895.

Sudhir, K.; Chou, T. M.; Chatterjee, K.; Smith, E. P.; Williams, T. C.; Kane, J. P.; Malloy, M. J.; Korach, K. S.; Rubanyi, G. M., Premature coronary artery disease associated with a disruptive mutation in the estrogen receptor gene in a man. Circulation 1997, 96, (10), 3774-7. 10.1161/01.cir.96.10.3774

Comment 4: Line 160. Rename the running title. The use of “transcriptomic” is misleading because only a set of genes were analyzed for the expression of transcripts.

Response: the title is now:

“2.6. Gene expression analysis of the main pathways involved in FMD”

Comment 5: Figure 7. Replace the value, 0.9198232 with 1.0 at the corresponding point in the expression spectrum.

Response: This is done.

Comment 6: Line183-185. Correct the duplicated entries.

Response: This is done.

Comment 7: Line 246. Spell out estradiol for E2.

Response: This is done.